# Development and Validation of the Military Minority Stress Scale

**DOI:** 10.3390/ijerph20126184

**Published:** 2023-06-20

**Authors:** Jeremy T. Goldbach, Sheree M. Schrager, Mary Rose Mamey, Cary Klemmer, Ian W. Holloway, Carl A. Castro

**Affiliations:** 1The Brown School, Washington University in St. Louis, St. Louis, MO 63130, USA; 2Department of Graduate Studies and Research, California State University, Dominguez Hills, Carson, CA 90747, USA; 3Amazon, Westborough, MA 01581, USA; 4Sexuality, Relationship, Gender Research Collective, University of Michigan, Ann Arbor, MI 48109, USA; 5Department of Social Welfare, Luskin School of Public Affairs, University of California Los Angeles, Los Angeles, CA 90095, USA; 6Suzanne Dworak-Peck School of Social Work, University of Southern California, Los Angeles, CA 90089, USA

**Keywords:** health disparities, LGBT, military, minority stress

## Abstract

Despite affecting nearly 3% of active-duty service members, little is known about how LGBT-related stress experiences may relate to health outcomes. Thus, the present study sought to create a Military Minority Stress Scale and assess its initial reliability and construct validity in a cross-sectional study of active-duty LGBT service members (*N* = 248). Associations between 47 candidate items and health outcomes of interest were analyzed to retain those with substantial betas. Item response theory analyzes, reliability testing, invariance testing, and exploratory factor analysis were performed. Construct validity of the final measure was assessed through associations between the sum score of the final measure and the health outcomes. The final 13-item measure demonstrated an excellent reliability (*ω* = 0.95). Bivariate linear regressions showed significant associations between the sum score of the measure and overall health (β = −0.26, *p* < 0.001), overall mental health (β = −0.34, *p* < 0.001), physical health (β = 0.45, *p* < 0.001), life satisfaction (β = −0.24, *p* < 0.001), anxiety (β = 0.34, *p* < 0.001), depressive symptoms (β = 0.37, *p* < 0.001), suicidality (β = 0.26, *p* < 0.001), and PTSD (β = 0.42, *p* < 0.001), respectively. This study provides the first evidence that minority stressors in the military setting can be operationalized and measured. They appear to have a role in the health of LGBT service members and may explain the continued health disparities experienced by this population. Little is known regarding the experiences of LGBT active-duty service members, including experiences of discrimination. Understanding these experiences and their associated health outcomes during military service may therefore help and guide further etiological studies and intervention development.

## 1. Introduction

Including both guard and reserve forces, nearly 71,000 (~2.8%) military personnel across all services identify as lesbian, gay, bisexual, or transgender [1,2] (LGBT). LGBT individuals have always served in the military, but until 2011, same-sex sexual behavior has been grounds for dismissal [3]. The repeal of the Don’t Ask, Don’t Tell, and Don’t Pursue policy (DADT) in 2011 lifted this ban, thereby allowing LGBT service members to be open about their sexual identities [4]. Although LGBT service members can no longer be involuntarily separated from the military, there has been an increased ambiguity regarding the military status of transgender service members. That is, transgender service members currently may continue to serve if they have either completed or begun the transition process, whereas those who have not begun will be dismissed if they come out as transgender.

Much of the existing knowledge regarding service members have been provided by veterans, as participation in research involving active-duty service members could jeopardize anonymity and thereby lead to dismissal [5]. Studies that have been conducted since the repeal of the DADT suggest that LGBT service members have continued concerns over discrimination, a lack of acceptance by unit leaders and fellow service members, and adverse effects on their military careers if they reveal their LGBT identity [6]. In the civilian literature, these types of fears fit in the paradigm of the minority stress theory (MST) [7]. MST suggests that as major life events and chronic discriminatory experiences accumulate, an individual becomes less equipped to adapt, adjust, and tolerate their continued life stressors [7,8]. The key stressors experienced by LGBT civilians that can lead to poor behavioral health outcomes include negative events (e.g., bullying and physical assault), negative attitudes regarding sexual and gender minority people (e.g., homophobia and transphobia), and discomfort with one’s identity (e.g., internalized stigma) and the coming-out process [9,10,11,12], all of which can also occur in a military context. However, no study to date has examined what minority stress looks like in the context of the military, or the prevalence of these minority stressors for individual service members.

The presence of minority stress Is also highly relevant to behavioral health patterns, and thus, military readiness. In civilian studies, LGBT individuals consistently demonstrate an increased stress and psychological vulnerability when compared to their non-LGBT peers [7,13]. Specifically, LGBT civilians have higher rates of depression [14], anxiety [15], PTSD [16], and substance use and abuse, compared to their majority peers [15,17,18,19,20,21]. Of particular interest in recent years is the prevention of suicide among both active-duty and veteran personnel [22], who make up more than 20% of suicide deaths annually in the United States [23]. Since 2001, suicide rates among the active-duty military members have doubled [24]. Few studies have explored suicide risk among the LGBT service members [25], but the general population literature consistently suggests an increased risk [26]. Blosnich et al. [27], using data from the California Quality of Life survey, found no significant differences in the suicidal ideation in the past 12 months, or in the attempts between LGBT and heterosexual veterans. However, this same study found a three times higher odds of lifetime suicidal ideation among LGBT veterans when compared to their heterosexual counterparts. These outcomes in the general population studies have been largely attributed to the presence of minority stress yet remain to be explored in the active-duty military context.

What military-specific minority stressors exist, their prevalence, and their relation to the behavioral health of LGBT service members all remain unclear. In part, this is because studies of minority stress, even in the general population, have been fraught with poor psychometric measurements [10,28,29]. In a recent review by Morrison and colleagues [30] of psychometric measurements assessing discrimination against sexual minorities, nearly all 162 articles reported the use of measures with sub-optimal psychometric properties. Common weaknesses included failure to examine scale dimensionality and poor assessments of content, criterion, and construct validity.

Thus, the present study sought to (a) identify military service-specific minority stressors through qualitative life history interviews (*N* = 42) with active-duty LGBT service members; (b) create a preliminary Military Minority Stress Scale (MMSS); and (c) refine the instrument, assess its reliability, and construct validity in a cross-sectional study of LGBT service members (*N* = 248). We also report on the rates of minority stress experiences that were reported by our sample. Based on the civilian literature, we hypothesized that unique military-specific minority stressors could be associated with an increased reporting of physical and behavioral health (e.g., depression, anxiety, alcohol and tobacco use, and suicidality) patterns.

## 2. Materials and Methods

The development and validation of the measure occurred in two phases, approved in advance by the Institutional Review Board. In Phase I, we developed candidate items through a qualitative inquiry with LGBT service members (*N* = 42). In Phase II, candidate items were included in a quantitative survey used to complete the development and validation process with a cohort of LGBT service members enrolled in a larger field study (*N* = 248). Methods and results for each phase of the research are described as follows.

### 2.1. Phase I: Participants and Sampling Procedures

The research team assembled an expert advisory panel of current and former military members known to the research team and through LGBT military networks. This panel met for a two-day in-person meeting, and this process informed the development of the recruitment plan and interview protocol. As LGBT service members can be difficult to reach [5], a multipronged recruitment strategy was employed: (a) a respondent-driven sampling method was used to take advantage of strong networks in this population; (b) to reach LGBT individuals who are not connected or “out” to others in the community, with the study advertising through each military branch’s official digital and print newspaper; and (c) the research team promoted the study in private Facebook groups for LGBT military personnel known to members of the research team. In an effort to ensure diversity of the sampling, as service members were enrolled, the research team monitored the racial, ethnic, service branch, sexual, and gender identity characteristics of all recruited participants. Nearing the end of study recruitment, the research team discontinued enrolling Air Force service members as they comprised more than 30% of the sample; however, no other groups were refused entry into the study. To participate in the interview, service members were required to (a) be at least 18 years of age; (b) speak English; (c) self-identify as lesbian, gay, bisexual, another sexual minority status, or transgender; (d) be active duty in the U.S. Air Force, Army, Marine Corps, or Navy; and (e) be willing and able to provide consent for their participation. 

### 2.2. Phase I: Data Collection Procedures and Instruments

Interested participants either emailed or called the research project office and were screened for eligibility. Research assistants were available during a 4 month period in 2016 to conduct participant interviews lasting approximately from 90 to 120 min. Interviews were conducted virtually using secure video-conferencing software at no cost to the participants. Participants had the option of communicating solely through the audio feature or using both the audio and video features. After reviewing the consent form and obtaining verbal agreement, participants completed audio-recorded interviews. Participants who were off duty when they participated received one $25 gift card and up to three $10 incentives for referring additional LGBT military members. 

Guided by the life history calendar method of interviewing participants [31], research assistants conducted semi-structured interviews, and allowed participants to identify salient experiences throughout their life as LGBT and in their military careers. Questions were consistent with the past use of life history calendar interviews with sexual minorities and included a discussion of life before and during military service [32], its relation to being “out”, and how these factors had influenced the participants’ experiences at work and with their friends and families. For example, questions included benchmarks regarding their experience before enlistment, during various stages of their careers (including during times of promotion), and the relationship between these milestones and their interpersonal relationships (e.g., “How did this influence your ability to date and find romantic partners?”) and structural factors (e.g., “How did the military support you during this transition, if at all?”). Four life history calendar interviews were initially conducted with members of the target population, with the researchers analyzing the data following each interview. Initial analyzes evaluated the procedures used by the interviewers and the applicability of the interview questions used with the specific aims of the study. 

### 2.3. Phase I: Analysis

All interviews were transcribed verbatim and entered into QSR NVivo. Informed by the relevant literature and theories (e.g., minority stress), the research team followed a thematic analytic process as outlined by Braun and Clarke [32] and developed a codebook to categorize data captured in respondent interviews. The research team edited the codebook as interviews were analyzed during a period of open coding, which included creating new codes to capture all distinct experiences and collapsing overlapping codes. For focused coding, research assistants paired up for three partnerships; each research assistant independently coded assigned interviews, which were co-coded by their partner. Partners used a coding consensus worksheet and discussion to reach a consensus; in situations where a consensus was not reached through discussion, coders then consulted a third party on the research team. The research coding team demonstrated a high degree of consistency between the study coders in use of the study codebook, with an interrater reliability of 96%. A more extensive description of the qualitative phase of this study can be found elsewhere [33].

### 2.4. Phase II: Participants

Participants for the quantitative survey were first recruited through referrals from the study’s expert advisory panel and study staff. Respondent-driven sampling was used to allow referred and eligible participants to act as seeds in the study [34]; seeds were asked to recruit up to three people in their network to build chains. Participants were also recruited through military-focused Facebook groups, newspapers, blogs, events, and conferences. Eligibility required participants to be aged 18 years or older; be currently active in the military (U.S. Army, Air Force, Marines, or the Navy); and to have reported a rank that aligns with their self-reported pay grade. Service members who were on duty at the time were allowed to take the survey they could participate but could not be compensated for participation. Eligible participants were then asked to give their consent and received information on the study accordingly. Two stages of fraud checks were used; the first attempted to remove people who were duplicates or fraud by checking (a) IP addresses that did not belong in the United States; (b) duplicate IP addresses or email addresses; and (c) similar data patterns between two participants’ response options. The second stage of fraudulent and validation checks were used after participants were deemed eligible for participation and assigned a participant ID number. Participants were flagged if they had more than one of the following: (a) a survey duration less than 20 min; (b) declined to answer more than 40 items in the survey (selected as “Decline to answer”); and (c) incorrect responses to two or more of the attention control questions in the survey (e.g., “Please select “No” as the answer for this question”). 

### 2.5. Phase II: Measures

#### 2.5.1. Demographic Characteristics

Demographics of interest included asserted gender; cisgender or transgender identity; sexual orientation; service branch; rank status; and race. Asserted gender was defined as male or female, wherein those who identified their gender as male or transgender male were coded as “male”, and those who identified their gender as female or transgender female were coded as “female”, respectively. Participants who identified as genderqueer or gender non-binary, or those who stated their gender identity as not listed in the response options provided, were not included in this recategorization (*n* = 7, 2.8%). Transgender status was defined as transgender or cisgender. Those who currently identified as a gender different than their sex assigned at birth or were non-binary were categorized as transgender, and those who identified as the gender they were assigned at birth were categorized as cisgender, respectively. Sexual identity was reported as heterosexual, gay or lesbian, bisexual, and other, respectively. Four service branches were included: the U.S. Army, U.S. Air Force, U.S. Marines, and U.S. Navy. Rank status was categorized as either officer or enlisted based on the choice of the participant. Race and ethnicity was recorded as Black or African American, Latino or Hispanic, White or Caucasian, Native American or Alaska Native, Asian or Pacific Islander, multiracial, and other race, respectively. Race and ethnicity was dichotomized for analysis as White or Caucasian vs. racial and ethnic minority due to small cell sizes in the Native American or Alaska Native, Asian or Pacific Islander, multiracial, and other race groups. Additional demographics that were included age and years of military service.

#### 2.5.2. Military Stress Experiences

The 47 candidate items for this scale were presented to those who identified as LGB, transgender, or both. The prompt asked participants to indicate whether each statement reflected their thoughts, feelings, and experiences since joining the military. Response options were “Yes”, “No”, or “Decline to answer”. Participants who selected “Yes” for a given statement were then presented with a follow-up question that asked whether that experience had happened to them in the past 30 days. The first set of questions asked about their overall perception of military policies and leadership (e.g., “I believe the military is unprepared to be inclusive to LGBT people”), leading up to more personal experiences, such as verbal harassment and physical assault. Participants were reminded that their responses were anonymous. 

#### 2.5.3. Physical and Mental Health Self-Assessment

A self-rated physical health question asked, “Overall, in the past 30 days, how would you rate your physical health?” Participants had response options of 1 (poor), 2 (fair), 3 (good), 4 (very good), 5 (excellent), and 6 (decline to answer). A similar question was asked for their overall past-30-day mental health with the same prompt and response options. 

#### 2.5.4. Physical Health

Physical health was assessed using the measures of somatic symptoms obtained from the Patient Health Questionnaire. This questionnaire identifies how much the person had been bothered by specific problems (e.g., back pain, stomach pain, and trouble sleeping) in the past 30 days. A three-point Likert-type scale was used to measure each question (0 = not bothered at all, 1 = bothered a little, 2 = bothered a lot). Scores ranged from 0 to 26 (theoretical range = 0–30), with higher scores indicating more problems with physical health (α = 0.88; 95% CI: 0.86–0.90).

#### 2.5.5. Alcohol Use

Given that substance use is strictly monitored in the military (with clear consequences), illicit drug use is uncommon and was therefore not assessed. Alcohol use was measured by the AUDIT-C [35], which involves summing the scores of three questions. Participants were first asked about their frequency of alcohol use (“How often do you have a drink containing alcohol?”), with response options of 0 (never), 1 (monthly or less), 2 (2–4 times a month), 3 (2–3 times a week), and 4 (4 or more times a week). Those who have ever drank alcohol were then given a follow-up question asking how many standard drinks containing alcohol do they have on a typical day when drinking (0 = 1 or 2, 1 = 3 or 4; 2 = 5 or 6; 3 = 7 to 9, and 4 = 10 or more, respectively). Finally, participants were asked how often they had six or more drinks on one occasion (0 = never, 1 = less than monthly, 2 = monthly; 3 = weekly, and 4 = daily or almost daily, respectively). The theoretical range of the sum scores was between 0 and 12, respectively (α = 0.63; 95% CI: 0.53–0.71).

#### 2.5.6. Life Satisfaction

The five-item Satisfaction with Life Scale was used to assess life satisfaction [36]. Participants indicated their level of agreement with each of the items: “In most ways my life is close to my ideal”; “The conditions of my life are excellent”; “I am satisfied with my life”; “So far I have gotten the important things I want in life”; “and “If I could live my life over, I would change almost nothing.” They used a seven-point Likert-type scale ranging from 1 (strongly disagree) to 7 (strongly agree). The total score was calculated by summing the responses (theoretical range: 5–35; α = 0.91; 95% CI: 0.89–0.93). 

#### 2.5.7. Anxiety Symptoms

The Generalized Anxiety Disorder scale was used to measure anxiety [37]. The seven questions were asked to participants to determine how often they had been bothered by problems in the last 2 weeks: feeling nervous, anxious, or on edge; not being able to stop or control worrying; worrying too much about different things; trouble relaxing; being so restless that it is hard to sit still; becoming easily annoyed or irritable; and feeling afraid as if something awful might happen. Scores for each item ranged between 0 (not at all) and 3 (nearly every day), with a total possible sum score ranging between 0 and 21, respectively (α = 0.95; 95% CI: 0.94–0.96).

#### 2.5.8. Depressive Symptoms

Depression was measured using eight of the nine items from the Patient Health Questionnaire [38]. The prompt asked participants to respond to questions about how often they have been bothered by depressive symptoms in the past 2 weeks. Response options for each item included 0 (not at all), 1 (several days), 2 (more than half the days), 3 (nearly every day), and “Decline to answer”. Sum scores were calculated for each participant. As the last item of the questionnaire (“Thoughts that you would be better off dead, or thoughts of hurting yourself in some way”) was not included in the survey, the theoretical range for depressive symptoms was found to be from 0 to 24, respectively (α = 0.93; 95% CI: 0.92–0.94). 

#### 2.5.9. Suicidality

Suicidality was measured with four questions. Participants were first asked: “Have you ever thought about or attempted to kill yourself?”. Those who responded with anything other than “Never” then received follow-up questions about their frequency of suicidal thoughts (“How often have you thought about killing yourself in the past year?”), with response options ranging from “Never” (1) to “Very often (5 or more times; 5); disclosure of suicidality (“Have you ever told someone that you were going to commit suicide, or thought you might do it?”) with response options including “No” (1), “yes, at one time, but did not really want to die” (2), and “yes, more than once, but did not want to do it” (3); and likeliness of suicide attempt (“How likely is it that you will attempt suicide someday?”), with response options ranging from “No chance at all” (1) to “Very likely” (6). Score were summed, with a theoretical range between 1 and 17 (α = 0.74; 95% CI: 0.66–0.81), respectively. 

#### 2.5.10. PTSD Symptoms

Total score on the PTSD Checklist for DSM-5 was used to assess PTSD symptoms [39]. This 20-item measure asked participants about experiences they had in the last 30 days. Response options ranged from 0 (not at all) to 4 (extremely), respectively, with higher scores indicating more extensive PTSD symptoms (theoretical range = 0–80; α = 0.97; 95% CI: 0.97–0.98).

#### 2.5.11. Perceived Acceptance

This study included two perceived acceptance questions: “On a scale of 0–100 (with 0 being the lowest and 100 being the highest), how accepted do you think the following people are within the military?” LBG service members and transgender service members were then referenced.

### 2.6. Phase II: Analysis

Each of the nine health outcomes of interest were first regressed onto the individual military stress items to assess the strength of these relationships. A meaningful association (β ≥ 0.10, corresponding to at least a small effect size, i.e., Cohen’s *d* > 0.20) between an item and an outcome were scored as 1, and otherwise were scored as 0. A sum score was then created for each item, wherein higher scores suggested that the item was meaningfully associated with more of the outcomes (theoretical range between 0 and 9, respectively). Items with eight or nine meaningful associations were then retained for additional analysis.

Next, item response theory (IRT) techniques were used to assess the difficulty and discrimination parameters of the retained items. The difficulty parameter represents how easily a person endorses (selects “yes”) a question, and the discrimination parameter estimates the degree to which a single item is measuring the same construct as the other items. To ensure that the development of a single measure could be used uniformly with active-duty service members, IRT was also employed to assess the test measurement invariance across the subgroups of interest, including asserted gender (male or female); rank (officer or enlisted); race and ethnicity (White or non-White); and gender minority status (transgender or cisgender). Unidimensionality was assessed by running an exploratory factor analysis on the candidate measure and assessing both eigenvalues (>1.00) and the scree plot. Cronbach’s alpha and composite reliability scores were assessed to measure their reliability. To ensure that the new composite measure retained its significant associations with the nine health outcomes, regression analyzes were again conducted on the sum score of the new measure. The *p*-values for these analyzes were adjusted using Benjamini and Hochberg’s procedure to minimize the likelihood of Type I errors [40]. All analyzes were performed SPSS versions 24(SPSS, Chicago, IL, USA) and 25 and Mplus 7.1 software (Mplus, Houston, TX, USA).

## 3. Results

### 3.1. Phase I

Demographics of the Phase I sample are included in Table 1. Of the 42 LGBT service members interviewed, 37 provided demographic information. The majority had an asserted gender of male (*n* = 24, 57.1%), identified their sexual orientation as gay or lesbian (*n* = 26, 61.9%), reported White racial or ethnic identity (*n* = 27, 6%), and were between 26 and 30 years of age (*n* = 16, 38.1%). Twelve participants each served in the Air Force (28.6%) and the Army (28.6%). Transgender service members comprised 21.4% of interview respondents (*n* = 9).

Eleven conceptual categories (e.g., military- and nonmilitary-related sexual minority stress, military identity, gender identity, sexual identity, military stress, general stress, military culture, health, and coping) were used to organize common experiences observed in the sample. Common types of experiences of minority stress related to military service and sexual orientation, including hiding identity status to avoid discrimination, and feeling left out of activities because of sexual or gender identity were found and subsequently coded. After the analysis, 47 candidate items from the study codebook were identified. These items were brought to the advisory board and the study team to be revised into close-ended statements. The final set of candidate items included in the Phase II survey are presented in Table 2.

### 3.2. Phase II

For Phase II, the sample consisted of 248 military personnel who identified as LGB, transgender, or both. Demographics for this group are presented alongside the Phase I participants in Table 1. Most military personnel were from the U.S. Army (*n* = 105, 42.3%), followed by the U.S. Air Force (*n* = 71, 28.6%), U.S. Navy (*n* = 50, 20.2%), and U.S. Marine Corps (*n* = 22, 8.9%), respectively. Slightly more than half of the sample identified as male (*n* = 142, 57.3%), followed by female (*n* = 99, 39.9%). Fifty-eight participants (23.4%) reported a transgender identity, wherein their sex assigned at birth was different than the gender with which they currently identify. The sample was majority White or Caucasian (*n* = 164, 66.1%), was enlisted (*n* = 148, 59.7%), and had an average age of 29 years (*SD* = 6.49). 

Frequencies for the 47 candidate military stress items are presented in Table 2. The most commonly endorsed experiences were “I am concerned that the military will change their policies to discriminate against LGBT service members” (*n* = 179, 72%); “Members of my unit have made negative comments about LGBT people” (*n* = 157, 63%); and “I know an LGBT service member who has been verbally harassed by other members of the military” (*n* = 149, 60%). 

Bivariate regressions were used to examine the relationship between the 47 candidate military minority stress items and each health outcome. Fourteen items had meaningful associations with at least eight of the nine measures and were selected as the top performers. A pilot measure was created featuring these 14 items and moved forward to the IRT analysis. 

The IRT analysis first examined the difficulty and discrimination parameters of these 14 items. These values assess the underlying experiences of the items and the degree to which they can be differentiated from one other. All items demonstrated acceptable difficulty values (range = 1.45–4.08) and discrimination values (range = −0.41–1.60). Figure 1 provides the item characteristic curve for this analysis. Based on these results, no items were removed at this stage.

Omega (composite reliability) scores were next calculated for the candidate measure. The 14-item measure had a high omega score [41] (*ω* = 0.95). After examining the scale with each item deleted, the removal of the item “I am concerned that the military will change their policies to discriminate against LGBT service members” increased the alpha to 0.887. This 13-item measure also produced an excellent composite reliability coefficient (0.951) and was chosen to move forward.

Finally, measurement invariance of the 13-item measure was tested across groups. For each invariance test, the loadings and thresholds of the configural (nonrestrictive) model were freely estimated and then constrained to equal in the scalar (restrictive) model. A decrease in the comparative fit index (CFI) greater than 0.01 was used to determine whether a decrement in fit existed between the configural and scalar models. Configural and scalar invariance was demonstrated for gender (male vs. female, ΔCFI = 0.015); transgender status (cisgender vs. transgender, ΔCFI = 0); race and ethnicity (White or Caucasian vs. non-White or non-Caucasian, ΔCFI = 0.001); and rank status (officer vs. enlisted, ΔCFI = 0.001). Thus, no items were excluded based on violating invariance.

To confirm the unidimensionality of the underlying factor structure, factor analysis was used with the principal components analysis (PCA) extraction method due to the dichotomous nature of the items and to inform the reduction of dimensionality by retaining the most informative variables [42,43]. A direct oblimin rotation was examined to allow for correlations among items. With the eigenvalue set at 1.00, three components emerged from the 13-item measure, with eigenvalues of 5.676, 1.103, and 1.003. Because two of these values hovered only slightly above the cutoff, we assessed the scree plot for guidance. The curve dramatically leveled off after only one component was plotted, confirming a single-factor measure. 

Sum scores for the 13-item military stress measure were created. Idiographic mean substitution was used for participants missing three or fewer items. Those missing more than three items were removed from this analysis, resulting in the retention of 239 participants for the measure validation analyses. Scores on the new measure ranged between 0 (no items endorsed) and 13 (all items endorsed) with a mean of 4.00 (*SD* = 3.78), suggesting that participants experienced, on average, mild to moderate military-related stressors. Finally, linear regressions were used to establish construct validity by assessing the hypothesized relationship between the MMSS score and each health outcome. As anticipated, findings showed that higher levels of minority stress were significantly associated with lower levels of overall physical health (β = −0.26, *p* < 0.001), mental health (β = −0.34, *p* < 0.001), and life satisfaction (β = −0.24, *p* < 0.001). Higher scores on the MMSS were associated with more physical health problems (β = 0.45, *p* < 0.001), anxiety (β = 0.34, *p* < 0.001), depressive symptoms (β = 0.37, *p* < 0.001), suicidality (β = 0.26, *p* < 0.001), and PTSD (β = 42, *p* < 0.001). Higher levels of minority stress were also associated with lower levels of perceived acceptance of both LGB service members (β = −0.37, *p* < 0.001) and transgender service members (β = −0.29, *p* < 0.001). There was no significant relationship between the MMSS and alcohol use. Table 3 presents these regression coefficients and their adjusted *p*-values.

## 4. Discussion

Although MST has provided a promising avenue to understand health disparities in civilian sexual and gender minority populations, to date we have been unable to examine this construct in the active-duty LGBT military population. This study represents the first data on minority stress experiences among active-duty LGBT service members. 

The Phase I interview study revealed common experiences of minority stress related to military service, including hiding identity status to avoid discrimination, and feeling left out of activities because of sexual or gender identity. Importantly, more than half of our sample reported experiencing stress related to several key areas, including concerns that the military will change their policies again to discriminate against LGBT service members, concerns with harassment that they have experienced themselves or seen others experience from both peers and leadership, a lack of military training on LGBT issues, and a lack of opportunity to build relationships with other LGBT service members. Given the shifts in policy at the highest levels of the department and executive leadership, these concerns are perhaps to be expected. However, better management of anti-LGBT harassment within units, an expansion of peer support, and better training for LGBT service members represent critical intervention points for consideration. 

These themes were expressed in 47 candidate military minority stress items, which were then assessed in the Phase II survey study. In this phase, the candidate itemset gave way to a final measure of 13 items that demonstrated excellent psychometric properties, including measurement invariance by key demographic subgroups and strong reliability. This 13-item MMSS was found to be significantly associated with all the health outcomes except for alcohol use. Relationships included those in minority stress and overall physical and mental health, life satisfaction, anxiety, depressive symptoms, suicidality, PTSD, and reporting of acceptance. Indeed, given that prior research has found increased rates in many of these outcomes among LGBT service members, our study further helps by explaining what may be driving these poorer behavioral health patterns, that is, our study helps move beyond simply documenting a relationship between one’s sexual and gender identity and health and begins to push towards points of intervention. This study provides the first evidence that minority stressors play an important role in the health of LGBT service members and may explain the continued health disparities experienced by this population.

Notably, the measure was not found to be associated with alcohol use, even after adjusting for multiple comparisons. In the military, alcohol consumption is extremely prevalent. In the present study alcohol use was high for both LGBT and non-LGBT service member. Thus, a lack of association between the MMSS and alcohol use in the present study was likely due to a ceiling effect and should therefore be explored further in subsequent research with this population.

### 4.1. Implications

In addition to its utility in field research, unit commanders can use the final 13-item measure as part of the command climate survey to assess a unit’s acceptance and integration of LGBT service members. Our results confirmed our hypothesized association between greater reports of minority stress and a lower perceived acceptance of both LGB and transgender service members, suggesting that the MMSS accurately captures the perceptions of bias against these groups within the unit. Importantly, as the MMSS was constructed to ensure measurement invariance across key demographic groups, it can be appropriately used regardless of a service member’s specific identities. That is, even if the mean levels of minority stress differ between identity groups, we would expect men to interpret these items and respond to them similarly to women. Likewise, we expect the measure to function equally well for both transgender and cisgender service members, for officers and enlisted service members, and so on. 

Examining the list of more common minority stress experiences may also provide a useful set of intervention targets for unit commanders and the department at large. In the present study, the most reported stressors included concern regarding the discriminatory policies, negative comments voiced by other service members in the unit, and verbal harassment. For instance, the military conducts annual psychoeducation training focused on treating all service members with dignity and respect. Based on the findings of this study, this training and similar trainings should be revised to highlight how negative comments voiced by other unit members and verbal harassment negatively affects the service member’s health and unit readiness.

### 4.2. Limitations

The present study has several limitations. First, the survey assessed only self-reported attitudes and behaviors, which can lead to either an over or underestimation of endorsed responses due to social desirability and other biases. In particular, service members may have underreported certain stressor items, given both the severity of these questions and because of the loyalty they may feel to their unit, leaders, and branch. Second, although the study gathered information from participants in the four main branches of the U.S. military serving across the country, our sample was not collected as a representative national sample, and the generalizability of our findings may be somewhat limited. Future studies would benefit from sampling from a larger set of service members across branches, deployment status, location (and urbanicity) of military installment, and other factors which may influence the daily experiences of LGBT service members. Similarly, given the smaller sample size, we cannot assert that our findings are equivalent across race and ethnicity, gender identity, age, and other intersectional factors. Related to this, participants in this study included both gender minorities and sexual minorities, which are two distinct groups that may experience different stressors and a varying severity of these stressors. Although this study had a sufficiently large sample to explore subgroup differences among sexual minority service members, the small group of individuals who identified as genderqueer or non-binary precluded further investigation into this subgroup, and invariance tests could not be performed to compare the experiences of transgender men to those of transgender women. The sample size also precluded splitting the dataset into training and validation samples. We recommend that obtaining a larger sample for confirmatory psychometric analysis is an important next step in this line of inquiry.

## 5. Conclusions

The measure developed in this study offers a robust and relatively short and easy-to-use tool for commanders and leaders to assess the acceptance and integration of sexual and gender minority service members. Future studies can use the MMSS to develop a comprehensive model of minority stress processes in the military and assess the effectiveness of anti-discrimination interventions. In addition, the military can use the MMSS in the Department of Defense’s biannual assessment of service member health and well-being to benchmark whether the department is making progress in LGBT acceptance and integration, including the assessment of the impact of LGBT status on health and readiness. 

## Figures and Tables

**Figure 1 ijerph-20-06184-f001:**
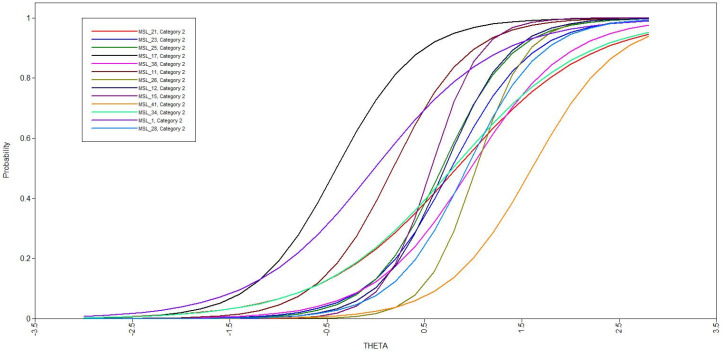
Item characteristic curves from the overall item response theory analysis with 14 items.

**Table 1 ijerph-20-06184-t001:** Frequencies of participant demographics.

	Phase I	Phase II
	*n* (%)	*n* (%)
*Asserted gender*		
Female	13 (31.0)	99 (39.9)
Male	24 (57.1)	142 (57.3)
Genderqueer or other gender	0 (0)	7 (2.8)
*Transgender*		
Transgender	9 (21.4)	58 (23.4)
Cisgender	28 (66.7)	190 (76.6)
*Sexual identity*		
Heterosexual or straight	3 (7.1)	20 (8.1)
Gay or lesbian	26 (61.9)	174 (70.2)
Bisexual	6 (14.3)	43 (17.3)
Other	2 (4.8)	11 (4.4)
*Service branch*		
U.S. Air Force	12 (28.6)	71 (28.6)
U.S. Army	12 (28.6)	105 (42.3)
U.S. Marines	3 (7.1)	22 (8.9)
U.S. Navy	10 (23.8)	50 (20.2)
*Rank status*		
Enlisted	--	148 (59.7)
Officer	--	100 (40.3)
*Race and ethnicity*		
White or Caucasian	27 (64.3)	164 (66.1)
Latino or Hispanic	5 (11.9)	33 (13.3)
Black or African American	3 (7.1)	20 (8.1)
Asian or Pacific Islander	2 (4.8)	13 (5.2)
Native American or Alaska Native	0 (0)	3 (1.2)
Multiracial or other	0 (0)	15 (6.0)
		*M* (*SD*)
*Age*	--	29.0 (6.5)
*Years enlisted*	--	7.5 (5.7)

**Table 2 ijerph-20-06184-t002:** Frequencies of responses for the initial 47 candidate items.

Item	*n* (%)
I am concerned that the military will change their policies to discriminate against LGBT service members.	179 (72.2)
* Members of my unit have made negative comments about LGBT people.	157 (63.3)
I know an LGBT service member who has been verbally harassed by other members of the military.	149 (60.1)
Required military trainings do not acknowledge LGBT issues.	145 (58.5)
There is a lack of opportunity to interact with other LGBT people in the military.	139 (56.0)
Military leadership has made negative comments about LGBT people.	138 (55.6)
I feel like the “unofficial representative” of LGBT people for my unit.	129 (52.0)
* I believe my leadership does not understand the needs of LGBT people.	121 (48.8)
My military healthcare provider is not trained to meet the needs of LGBT people.	120 (48.4)
I believe military policies are unsupportive of LGBT service members.	116 (46.8)
There are no role models for LGBT people in the military.	113 (45.6)
Members of my unit have told me that being LGBT is against their religion.	111 (44.8)
* I believe some members of my unit do not respect me because I am LGBT.	107 (43.1)
I believe there are fewer military benefits for LGBT service members than non-LGBT service members.	107 (43.1)
I believe members of my unit do not want to work with LGBT service members.	105 (42.3)
I feel isolated from other LGBT people because I am in the military.	103 (41.5)
I need to hide that I am LGBT to avoid it negatively affecting my career.	101 (40.7)
It is acceptable to make anti-LGBT statements in my unit.	96 (38.7)
I was outed to someone in my unit without my permission.	96 (38.7)
I believe the military is unprepared to be inclusive to LGBT people.	96 (38.7)
I need to hide that I am LGBT in order to be accepted by members of my unit.	95 (38.3)
I am concerned about being treated unfairly by my supervisors if they find out I am LGBT.	93 (37.5)
I feel pressure because I believe I am the only openly LGBT person in my unit.	91 (36.7)
I believe my performance is held to a different standard than my military peers because I am LGBT.	87 (35.1)
I am concerned I will lose my job/rank because I am LGBT.	79 (31.9)
* Someone in the military has verbally harassed me for being LGBT.	71 (28.6)
* My supervisor has made negative comments about LGBT people.	70 (28.2)
* Some members of my unit are unwilling to acknowledge my LGBT identity.	70 (28.2)
* I have been rejected by members of my unit when they found out I am LGBT.	70 (28.2)
I know an LGBT service member who has been physically assaulted by other members of the military.	65 (26.2)
* I believe members of my unit exclude me from unit activities because I am LGBT.	65 (26.2)
* I don’t fit into military culture because I am LGBT.	63 (25.4)
Members of my unit have told me to hide that I am LGBT.	55 (22.2)
* In my unit, it would be pointless to report verbal harassment against LGBT service members.	53 (21.4)
The military’s LGBT policies are not followed/supported in my unit.	53 (21.4)
* Being LGBT has negatively affected my ability to advance in my military career.	52 (21.0)
I have concerns about my physical safety in the military because I am LGBT.	51 (20.6)
I feel rejected by the LGBT community because of my military service.	44 (17.7)
* I have been treated unfairly by my supervisors when they found out I am LGBT.	40 (16.1)
My supervisor has told me to hide that I am LGBT.	30 (12.1)
* In my unit, it would be pointless to report physical violence against LGBT service members.	23 (9.3)
I was forced to out myself because I got “caught”.	23 (9.3)
Other members in my unit have retaliated against me for reporting verbal harassment against LGBT service members.	22 (8.9)
Someone in the military has physically assaulted me for being LGBT.	20 (8.1)
Other members in my unit have retaliated against me for reporting physical violence against LGBT service members.	11 (4.4)
Leaders in my unit have retaliated against me for reporting verbal harassment against LGBT service members.	11 (4.4)
Leaders in my unit have retaliated against me for reporting physical violence against LGBT service members.	8 (3.2)
* Item retained in the final 13-item measure.	

**Table 3 ijerph-20-06184-t003:** Regressions of 13-Item Sum Score and Health Outcomes.

Outcome	Beta	*p*-Value	F-Statistic	df_num_	df_den_	R^2^
Physical Health Self-Assessment	−0.26	<0.001	17.21	1	247	0.07
Mental Health Self-Assessment	−0.34	<0.001	31.00	1	247	0.14
Physical Health	0.45	<0.001	59.04	1	247	0.23
Alcohol	0.12	0.102	2.712	1	199	0.02
Life Satisfaction	−0.24	<0.001	13.82	1	247	0.07
Anxiety	0.34	<0.001	27.98	1	232	0.12
Depressive Symptoms	0.37	<0.001	36.86	1	247	0.17
Suicidality	0.26	<0.001	16.80	1	247	0.08
PTSD	0.42	<0.001	50.77	1	247	0.20

## Data Availability

Data reports and datasets can be obtained by emailing the lead author and coordinating a data sharing agreement, Jeremy T. Goldbach (jgoldbach@wustl.edu).

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
