# Peer review of "Development and Validation of the Military Minority Stress Scale"

_ijerph, 2023, doi:10.3390/ijerph20126184_

Round 1

Reviewer 1 Report

The article grounds on a study which aimed to develop a scale measuring Military Minority Stress and assessing its validity and reliability.

The main question of the research is how to operationalize and measure stressors among LGBT people employed in the Military Service. The topic is original and relevant as a considerable proportion of military persons belongs to sexual minority groups and they still are affected by discrimination. As far as I know there is no proper tool for measuring the effects of this discrimination on mental and health status in military services. Previous studies examined only civilian sexual minority population; this study is the first one which covers active military service members.

The purpose is well supported by the authors, and they give a good overview of the relevant literature. References are all along appropriate.

As a method they use qualitative inquiries and a quantitative survey. We get a very systematic and precise description of both procedures, authors’ arguments for reasoning the selected tools and analyses are compelling. Each phase of the study is well considered and methodologically underpinned. The whole study design and its presentation are a merit of the article.

As a further step, the elaborated measuring toll should be tested on a bigger and representative sample.

The novelty of the study is that it provides a tool for measuring minority stress in military service. The validity of this tool has been proved by the analyses.

The conclusions are consistent with the evidence presented, they reflect to the main question of the study, but the discussion chapter also includes some substantive result related to the role of minority stressors in the health of LGBT service members.

Tables and figures are correctly presented.

The article presents a thoroughly elaborated well corroborated study.

Reviewer 2 Report

Development and Validation of the Military Minority Stress Scale

This is an interesting and important contribution to assessing minority scale in military setting in the USA. The article is well written with adjusted methodological approaches and decisions. Still, I believe a few changes in the article would improve the overall chances of it being published:

1.     Title. I believe the article would benefit from the inclusion of the word “sexual” minority stress scale to provide more clarity and accuracy to what is being measured.

2.     Please provide more details regarding the qualitative inquiry with LGBT service members, and examples of questions of the interview script.

3.     Please prove reliability scores for all measures of health used.

4.     I suggest table 2 to be included as a supplemental file and the final 13-item solution presented in the article.

5.     Authors should use some of the health measures used to present information regarding discriminant/convergent validity.

6.     Line 386: what does a mean score of 4.00 mean? Authors should state that this represents high levels of minority stress.

7.     The discussion section would benefit from a deeper understanding and relativization of these results. For example, why were all measures associated with the scale except alcohol? Shouldn’t the obtained score for anxiety be higher? And so on…

8.     Table 3 lacks more information from the regression made. Please provide information regarding the R2 value (the coefficient of determination), the F value (also referred to as the F statistic), the degrees of freedom in parentheses, and the p value.

9.     Please provide an implications section of these results.

Best wishes.

Reviewer 3 Report

The paper shows a model example of new measure creation. The initial stage (qualitative) was conducted very well (if I understand correctly, those results were already published). The second stage (quantitative) has also been thought through well. What could be improved is the sampling for the quantitative stage (which the authors indicate in the limitations themselves). The only other thing worth mentioning (and - maybe - controlling for ?) is the case of the overfitting of the model. The issue is that the authors take to their questionnaire only the items that have a high relation with the criterion variables (a good procedure in itself), but due to the fact that they use THE SAME sample to both create the questionnaire AND assess its validity, the results may be statistically boosted. What I would suggest (and the idea is not mine, but comes from many authors) is to FIRST split the sample into training sample (used to build the items, along with checking the relation with health outcomes), and a second, verification/test sample (which remains untouched in the first step). Then, after "creating" the item set on the first half sample (and I really applaud the use of IRT for that), it can be TESTED on the second "fresh" sample. There is a good chance the results will be the same, but it would secure that there is no statistical overfitting of the items.

As for all the other mathers, the article is good and can be published.
